# Does a Specific Sequential Combination of Antiseptic Solutions for Chemical Debridement in Periprosthetic Joint Infection Improve Outcomes vs. Solution Alone? An In Vivo Study

**DOI:** 10.3390/antibiotics13121225

**Published:** 2024-12-17

**Authors:** Miguel Márquez-Gómez, Marta Díaz-Navarro, Andrés Visedo, Lourdes Prats-Peinado, Patricia Muñoz, Javier Vaquero, María Guembe, Pablo Sanz-Ruíz

**Affiliations:** 1Department of Orthopaedic Surgery and Traumatology, Hospital General Universitario Gregorio Marañón, 28007 Madrid, Spain; miguel.marquez@salud.madrid.org (M.M.-G.); lourdes.prats@salud.madrid.org (L.P.-P.); javier.vaquero@salud.madrid.org (J.V.); pablo.sanz@salud.madrid.org (P.S.-R.); 2Instituto de Investigación Sanitaria Gregorio Marañón, 28009 Madrid, Spain; marta.diaz@iisgm.com (M.D.-N.); andres.visedo@iisgm.com (A.V.); pmunoz@hggm.es (P.M.); 3Department of Clinical Microbiology and Infectious Diseases, Hospital General Universitario Gregorio Marañón, 28007 Madrid, Spain; 4CIBER Enfermedades Respiratorias-CIBERES (CB06/06/0058), 28029 Madrid, Spain; 5Medicine Department, School of Medicine, Universidad Complutense de Madrid, 28040 Madrid, Spain

**Keywords:** chemical debridement, sequential combination, reduction, biofilm, murine model

## Abstract

**Background**: Chemical debridement is a fundamental step during the surgical treatment of both acute and chronic periprosthetic joint infection (PJI). However, there is no consensus on the optimal solution, nor is there sufficient evidence on the optimal irrigation time and combination of solutions. In an in vitro study, our group recently demonstrated that sequential combination debridement (SCD) with 3% acetic acid (AA) followed by 10% povidone iodine (PI) and 5 mM hydrogen peroxide (H_2_O_2_) was the best strategy for reducing bacterial load. The present study aimed to validate these findings in an in vivo model. **Results**: The median (IQR) log CFU/mL was lower in the group of mice treated with SCD (2.85 [0.00–3.72]) than in the Bactisure™ group (4.02 [3.41–4.72], *p* = 0.02). While this reduction was also greater than in the PI group (3.99 [1.11–4.33]), the difference did not reach statistical significance (*p* = 0.19). Cell viability assays showed no differences between treatments. *S. aureus* bacteremia was detected in 10% of mice treated with SCD, compared to 30% in the PI group and 10% in the Bactisure™ group. The difference was not statistically significant (*p* = 0.36). **Conclusion**: Our findings confirm that SCD significantly reduced bacterial load in an in vivo S. aureus PJI model, showing superior anti-biofilm activity compared to Bactisure™ and comparable performance to PI alone. These results highlight SCD’s potential to serve as a standardized chemical debridement protocol, combining enhanced efficacy with clinical applicability. **Methods**: We tested SCD with 3% AA for 3 min, 10% PI for 3 min, and H_2_O_2_ for 3 min in a 7-day *Staphylococcus aureus* (ATCC29213)-based murine femur PJI model and compared the results with single treatments of 10% PI for 3 min or Bactisure™ solution for 3 min. A sterile steel implant with local administration of saline solution for 3 min was used as a non-infected control. After completing irrigation procedures, under anesthesia, mice were euthanized, and implants were analyzed for CFU/mL counts and cell viability rates. Blood cultures were obtained pre-euthanasia to detect bacteremia.

## 1. Introduction

Periprosthetic joint infection (PJI) is a severe and challenging complication of joint arthroplasty, with an incidence of 1–3% in primary surgeries and up to 10% in revision procedures [1]. This complication has a significant impact on patients’ quality of life, often necessitating multiple surgical procedures, prolonged hospital stays, and extended antibiotic therapy, while also imposing a substantial economic burden on the healthcare system [2,3].

Managing PJI is complicated by the presence of bacterial biofilms on the implant surface. These are highly resistant to both antibiotics and the host immune system [4]. Biofilms are structured microbial communities encased in a self-produced extracellular matrix that adheres to surfaces, significantly enhancing bacterial resilience. Primary treatment options for PJI include surgical debridement with implant retention, one- or two-stage revision surgeries, and prolonged antibiotic therapy [5,6]. However, recurrence rates remain high [7]. Surgical, mechanical, and chemical debridement are key components of PJI surgery aimed at reducing bacterial load, removing existing biofilm, and preventing reinfection [8].

The use of antiseptic solutions for chemical debridement has been an integral part of the management of PJI, yet there is no consensus on the optimal solution or duration of irrigation [9]. Antiseptics such as povidone iodine (PI), hydrogen peroxide (H_2_O_2_), and 3% acetic acid (AA) are widely used, although studies comparing their efficacy are limited. Short-term biofilm models, such as the 24 h biofilm system, are commonly employed to assess early biofilm formation and treatment efficacy. These models are pivotal for understanding the behavior of immature biofilms and optimizing antiseptic protocols. In vitro studies have shown that PI, particularly at a concentration of 10%, exhibits strong bactericidal activity against biofilm-forming organisms such as *Staphylococcus aureus* and *Pseudomonas aeruginosa* when applied for at least 3 min [10]. Recent evidence suggests that combining antiseptic solutions such as H_2_O_2_ and PI sequentially may have synergistic antibacterial effects, especially against mature biofilms [10,11].

In a recent in vitro study, our research group demonstrated that sequential combination debridement (SCD) with 3% AA followed by 10% PI and 5 mM H_2_O_2_ (3 min each) was the most effective strategy for reducing bacterial biofilms on steel implants [12]. This sequential combination works by first using AA to disrupt the biofilm matrix, followed by the application of PI for its potent bactericidal action and H_2_O_2_ to increase cell wall porosity and enhance bacterial killing. However, these results have not yet been validated in vivo. Therefore, this study aimed to evaluate the anti-biofilm efficacy of this antiseptic combination using a modified murine model of osteoarticular infection.

## 2. Results

### 2.1. Physical Appearance and Clinical Data

The mean (±SD) age of the mice was 10.63 ± 0.95 weeks. The mean (±SD) weight before surgery (day 0) was 25.4 ± 2.31 g. At day 6 of observation, the mean (±SD) weight decreased to 23.63 ± 2.52 g. The difference was statistically significant (paired *t*-test, *p* < 0.001) (Table 1, Figure 1).

No statistically significant differences were found for weight loss between the control group (with non-infected implants) and the treatment groups (*p* = 0.52).

Only two of the six behavioral variables assessed—limping and piloerection—were affected during the observation period as a postoperative measure of pain. None of the mice exhibited a lack of grooming, wounds, passivity, or aggression. On the first day after surgery, all the mice demonstrated signs of limping, restricted movement, and avoidance of weight-bearing on the affected limb. This percentage decreased progressively to 30% on the second day and 12.5% on the third day. By the fourth day, none of the animals were limping. Piloerection was noted in 37.5% of the mice on the first postoperative day, followed by a decrease on the second and third days, and had completely resolved by the fourth day (Appendix A).

None of the mice in the non-infected control group exhibited wound dehiscence or signs of infection, such as erythema, swelling, purulent exudate, abscesses, or fistulas. In the treatment groups, no wound dehiscence was observed, although all the mice displayed clear signs of infection, including swelling and abscess formation around the knee joint. No animals were lost owing to infection during the 7-day observational period. These findings were confirmed through dissection and macroscopic examination following euthanasia, 1 week after implant surgery (Figure 2).

### 2.2. Microbiological Analysis

All the samples obtained from the sonication of the implants in the 10 mice from the negative control group were sterile, as expected.

In the treatment groups, the median (IQR) log CFU/mL of the sonicated product was 3.99 (1.11–4.33) for the implants treated with 10% PI, 4.02 (3.41–4.72) for the implants treated with Bactisure™, and 2.85 (0.00–3.72) for the implants treated with SCD (Table 2).

The median (IQR) log CFU/mL obtained from the sonicated implants was lower in the group of mice treated with SCD (2.85 [0.00–3.72]) than in those treated with Bactisure™ (4.02 [3.41–4.72], *p* = 0.02; 29.10% reduction). No significant differences were found in the median (IQR) log CFU/mL between mice treated with 10% PI (3.99 [1.11–4.33]) and those treated with SCD (2.85 [0.00–3.72], *p* = 0.19) or Bactisure™ (4.02 [3.41–4.72], *p* > 0.99) (Table 3) (Figure 3).

No mice in the control group developed bacteremia, defined as the growth of *S. aureus* in blood culture. Bacteremia was observed in three mice (30%) from the 10% PI group, and in only one mouse (10%) from the Bactisure™ and SCD groups. The differences between the groups were not statistically significant (Fisher’s exact test, *p* = 0.36) (Table 4).

### 2.3. Cell Viability Assays

The median (IQR) cell viabilities for PI10, Bactisure™, and SCD were, respectively, 31.61 (23.80–55.06), 24.00 (16.18–37.37), and 31.93 (23.75–46.24). No significant difference in cell viability was found between the three treatments (*p* = 0.34) (Figure 4). These results indicate that all three treatments exhibited similar cytotoxic effects on cell viability in this in vivo experiment.

## 3. Methods

### 3.1. Setting

The study was performed in the microbiology laboratory and the animal facility of a tertiary institution in Madrid, Spain, in accordance with the principles of the 1964 Declaration of Helsinki (as revised in 2013). The study protocol was approved by the Research Ethics Committee of our center. The project has the registration code ES280790000087 and the reference PROEX 022.1/24.

### 3.2. Ethics Approval and Animal Handling

This study was approved by the Dirección General de Agricultura, Ganadería y Alimentación, and the Consejería de Medio Ambiente, Vivienda y Agricultura (PROEX no. 022.1/24). It was conducted in accordance with local legislation and institutional requirements, specifically with Royal Decree 53/2013 and Law 32/2007 regarding basic standards for the protection and care of animals used in experimentation.

The ethical principles of the 3 Rs (replacement, reduction, and refinement) were followed. Animals were handled according to the guidelines of Directive 2010/63/EU, Law 9/2003, and Royal Decree 178/2004 on the use of genetically modified organisms.

### 3.3. In Vivo Model

We designed a 7-day *S. aureus* (ATCC29213)-based murine PJI model to be run in biosafety level 2 of the animal housing facility of our institution. This model was based on a previously validated 24 h biofilm system [13] to represent the early stages of biofilm formation, which was adapted to simulate the progression of infection over a 7-day period for in vivo validation. The 7-day infection model was selected based on prior validated studies and the need to balance infection development with ethical considerations. The study sample comprised 40 BALB/c mice aged between 6 and 15 weeks. The animals were provided by the animal facility of the Experimental Medicine and Surgery Unit (MCEU) at General University Hospital Gregorio Marañón.

#### 3.3.1. Study Groups

Group 1 (control): Sterile steel implant with local administration of sterile saline solution for 3 min.Group 2 (PI10): Contaminated steel implant with local administration of 10% PI for 3 min.Group 3 (Bactisure™, Zimmer Biomet, Zug, Switzerland): Contaminated steel implant with local administration of Bactisure™ (ethanol [solvent], AA [pH modifier], sodium acetate [buffer], benzalkonium chloride [surfactant], and water) for 3 min.Group 4 (SCD): Contaminated steel implant with local administration of a combination of 3 solutions in the following order: 3% AA for 3 min, 10% PI for 3 min, and 5 mM H_2_O_2_ for 3 min. Saline washes were applied between dilutions.

#### 3.3.2. Implant Preparation

For the preparation of the implants, surgical steel Kirschner wires measuring 0.8 mm in diameter were measured and cut to obtain implants measuring 12 mm in length. The implants were then individually packaged and sterilized using an autoclave (121 °C, 15 min) before use.

All implants, except those in the control group, were contaminated for 24 h, at 37 °C, in an incubator with an orbital shaker and 1 mL of a 1 × 10^7^ CFU bacterial suspension of *S. aureus* in fresh tryptic soy broth. For the non-infected negative controls, the implants were kept in sterile tryptic soy broth for 24 h at 37 °C. After this period, the discs were washed 3 times with phosphate-buffered saline to remove non-adherent bacteria before insertion (Figure 5).

#### 3.3.3. Experimental Model

The surgery was performed in the animal housing and surgical areas of the MCEU at General University Hospital Gregorio Marañón.

After administering anesthesia (medetomidine hydrochloride 1 mg/kg subcutaneously and ketamine 150 mg/kg intraperitoneally) and appropriate analgesia, the skin of the left leg was shaved and prepared with alcoholic chlorhexidine. The animal was placed in the supine position, with both front legs and the right hind leg secured to the work surface, while the left hind leg was left free for easier hip and knee flexion. A longitudinal incision centered on the patellar tendon was made over the shaved skin, followed by a medial parapatellar arthrotomy to access the distal femur. The patella was luxated laterally, and a 25G needle was inserted into the femoral medullary canal. A 21G needle was introduced to widen the canal and thus enable retrograde placement of a 12 mm long, 0.8 mm diameter surgical steel implant, leaving 1 mm in contact with the knee joint. Finally, the extensor mechanism was reduced, and the arthrotomy and skin were closed using non-absorbable monofilament sutures. The main steps of the surgery are detailed in Figure 6.

The mice were housed on a ventilated rack in a biosafety level 2 animal facility for 7 days. Pain was managed with buprenorphine (0.1 mg/kg subcutaneously) and ibuprofen in their drinking water (1 mL/100 mL). Pain and the condition of the wound were monitored daily, while the weight of each animal was recorded every 48 h on weekdays to ensure their well-being. Pain was assessed based on the presence or absence of 6 behaviors directly related to pain or stress in this species and the surgical procedure they had undergone, namely, limping, piloerection, lack of grooming, presence of wounds, passivity, and aggression.

After 1 week, the mice underwent a second surgical intervention. The knee was re-exposed to reveal the implant, and approximately 2 mL of antiseptic solution was administered locally, ensuring the entire surgical area was covered. This procedure simulated the chemical debridement typically performed during PJI surgery. The surgical area was washed with sterile saline before a debridement sequence was performed, and the debridement solutions were then added sequentially. Between solutions and at the end, the area was rinsed again with saline.

After treatment and while still under anesthesia, the mice were euthanized by cervical dislocation, and the implants were removed for processing after being rinsed with sterile saline solution to eliminate non-adherent bacteria. At the same time, 0.3 mL of intracardiac blood was collected by direct puncture to assess the presence of bacteremia.

Sessile cell recovery: The implants were individually transferred to new glass tubes containing 1 mL of phosphate-buffered saline and sonicated for 10 min at 40 kHz to detach the biofilm. The sonicated bacterial suspension was then vortexed for culture.

Culture of sonicate: Once the sonicated bacterial suspension had been vortexed, one part was serially diluted, and 100 µL of the dilution was cultured on blood agar plates. The plates were incubated at 37 °C for 24 h, and CFU/plate counts were calculated and expressed as CFU/mL on a logarithmic scale.

Cell viability rate: The sonicated bacterial suspensions were refrigerated and further stained on slides with a commercial BacLight^®^ kit (ThermoFisher Scientific, Madrid, Spain) (comprising Syto 9, which stains living cells green, and propidium iodide, which stains damaged cells red), according to the manufacturer’s instructions. The samples were analyzed using flow cytometry (Gallios, Beckman Coulter, BioRad, Madrid, Spain) and images were acquired by adjusting the settings to include only singlets in the analysis. The resulting data were analyzed using the Kaluza software application (version 2.2).

Bacteremia: The presence of bacteria in the blood was assessed by incubating the intracardiac blood sample in an aerobic blood culture bottle for up to 5 days. If the culture was positive, a blood agar plate was used to confirm the qualitative presence of *S. aureus*.

#### 3.3.4. Statistical Analysis

Quantitative variables are expressed as the median and the interquartile range (IQR). The median (IQR) log CFU/mL and median (IQR) cell viability rates were calculated for each experiment. The median (IQR) values are detailed in the tables and figures.

Owing to the small sample size, we used nonparametric methods (Kruskal–Wallis test and Fisher’s exact test) for comparisons between groups. Linear or logistic regression models were fitted in cases of asymmetry.

Statistical significance was set at *p* < 0.05 for all the tests. The statistical analysis was performed using IBM SPSS Statistics for Windows, Version 21.0 (IBM Corp, Armonk, NY, USA), and Prism 10.0 (GraphPad, San Diego, CA, USA).

### 3.4. Availability of Data and Material

All the data generated or analyzed during this study are included in this published article and its Appendix A. Additional data are available from the corresponding author upon request.

## 4. Discussion

Chemical debridement is an important step in PJI surgery, regardless of the chronology of infection, as it reduces bacterial load, disrupts biofilm, and helps prevent reinfection. While there is general agreement on using antiseptic solutions for this purpose, the optimal solutions and application times remain controversial, with limited evidence available. This study aimed to validate our previous in vitro findings, which identified SCD using AA, followed by PI and H_2_O_2_, as the most effective strategy for reducing bacterial load and biofilm on steel implants. In a 7-day in vivo murine model of *S. aureus* PJI, we compared the efficacy of different chemical treatments, including 10% PI alone, Bactisure™, and SCD of 3% AA, 10% PI, and 5 mM H_2_O_2_.

All the mice showed significant weight loss, regardless of group, reflecting postoperative stress. Importantly, no statistical differences in weight loss were observed between the infected and non-infected groups (*p* = 0.52), suggesting that infection itself did not disproportionately contribute to systemic effects. Behavioral variables like limping and piloerection were observed transiently, resolving by day 4, further indicating effective recovery in our murine model. These findings support the model’s relevance and reproducibility in studying PJI.

Our results confirmed that SCD was the most effective approach for reducing bacterial load (CFU/mL) compared to Bactisure™ (*p* = 0.02) and 10% PI alone, although the difference for the latter was not statistically significant. These findings validate our previous in vitro results [12], in which this combination also showed the greatest reduction in biofilm on surgical steel, both in terms of CFU and cell viability. The sequential order of antiseptic application can influence bactericidal activity. As observed in our previous in vitro experiment [12], the most effective SCD treatments were those initiated with AA, a weak organic acid that disrupts the biofilm matrix, enhancing the bactericidal effects of subsequent agents like PI and H_2_O_2_.

Moreover, as a weak organic acid, AA’s diffusion capacity allows it to penetrate tissue regions that are inaccessible to mechanical debridement or other antiseptics with lower penetration abilities [14]. While AA is not primarily used for its antiseptic properties due to its prolonged action time (approximately 20 min) for achieving bactericidal effects [15], its role in combination therapy is critical. By breaking down the biofilm matrix, AA facilitates subsequent bactericidal action by PI, which directly targets bacteria, and H_2_O_2_, which increases cell wall porosity and enhances the effectiveness of PI due to their synergistic interaction. Notably, our findings suggest that the initial use of AA in the SCD sequence is the only step that contributes significantly to reducing biofilm viability, underlining its role in the sequence. This is relevant not only to orthopedic surgery, but also to any implant-related infection across surgical fields.

The lack of significant differences in reducing cell viability observed in this in vivo experiment aligns with our group’s previous in vitro findings [12], suggesting that while a reduction is present, it is not substantial. Nevertheless, our findings suggest a synergistic effect between the antiseptic solutions studied. Although SCD can reduce cell viability, the improvement is limited, and further studies are needed to explore agents that can more effectively target cell viability. This lack of correlation between the reduction in log CFU/mL and viable cells may be due to the presence of viable but non-culturable cells, and could also explain the differences observed, as we previously hypothesized in the case of breast prostheses [16]. The bacteremia observed immediately after debridement likely reflects a pre-existing infection, given the rapidity of the observed effect, and we did not find a difference between treatments.

Consistent with the literature, 10% PI exhibited strong anti-biofilm activity, as demonstrated in in vitro studies reporting a 95% reduction in biofilm [10,17]. However, concerns about the cytotoxicity of PI at this concentration persist, particularly in human fibroblasts, osteoblasts, and chondrocytes, as shown in in vitro studies [18,19]. The cytotoxicity of PI depends on concentration and application time, and while 10% PI is more toxic than the commonly used 0.3%, the toxicity concerns are mostly based on in vitro or prophylactic studies [20]. These results remain invalidated in the treatment of PJI, and further research is needed to assess in vivo risks at these concentrations.

According to the ICM 2018 guidelines [21], the optimal irrigation solution for clean elective orthopedic procedures is dilute PI, although the optimal volume remains undefined. However, no specific recommendation is made for irrigation during PJI surgery, due to insufficient evidence. While dilution with sterile saline can reduce bacterial load, antiseptics play a crucial role in eradicating infection by disrupting biofilm and creating a hostile environment that prevents future bacterial growth.

Bactisure™ was less effective than PI and SCD in our model, consistent with earlier studies showing its inferior performance against biofilm on arthroplasty surfaces, particularly PMMA cement and plastic [10]. Despite containing agents such as AA and ethanol, the ability of Bactisure™ to eradicate biofilm appears weaker than that of PI or SCD.

Our study was limited by the use of a 24 h biofilm maturation model, which may not fully represent mature biofilms in clinical PJIs. However, the model does reflect early biofilm formation, which plays a crucial role in infection and is a target in procedures like debridement, antibiotics, and implant retention (DAIR), aimed at eradicating infection without the revision of fixed components. Additionally, as this was an animal study, the sample size was limited by the reduction principle, and inter-animal variability was high due to host factors such as immune response and biofilm formation dynamics, which constrained treatment comparisons and statistical power. Surfactants and topical antibiotics were not evaluated, owing to the lack of evidence supporting their use in these settings [22]. Although our study focused on the most effective solutions identified in vitro—PI10, Bactisure™, and SCD—the results remain robust, with SCD showing the lowest CFU count and cell viability after sonication of the implant.

Our findings may have been influenced by the sample storage conditions, and our focus on surgical steel implants without considering other materials such as PMMA, titanium, or polyethylene, which are common in orthopedic devices. Biofilm growth can vary across surfaces, affecting the efficacy of the antiseptic. Our findings are specific to *Staphylococcus aureus* and cannot be directly extrapolated to other microorganisms or polymicrobial biofilms. Bacterial load in adjacent bone tissue was also not assessed, due to the short infection duration (7 days), which likely limited biofilm spread to surrounding tissues. Additionally, we did not evaluate the mechanical effects of pulsatile lavage. Future studies should address these variables in more comprehensive in vivo models, as well as the safety and long-term effects of debridement on infection control.

## 5. Conclusions

Based on our in vivo study, SCD with 3% AA, 10% PI, and 5 mM H_2_O_2_ demonstrated superior efficacy in reducing biofilm in a murine PJI model, compared to commercially available solutions like Bactisure. These findings validate our in vitro results, and suggest that this combination could serve as an effective chemical debridement strategy during PJI surgery. However, further clinical data are needed to confirm whether these solutions can effectively reduce PJI in clinical settings.

## Figures and Tables

**Figure 1 antibiotics-13-01225-f001:**
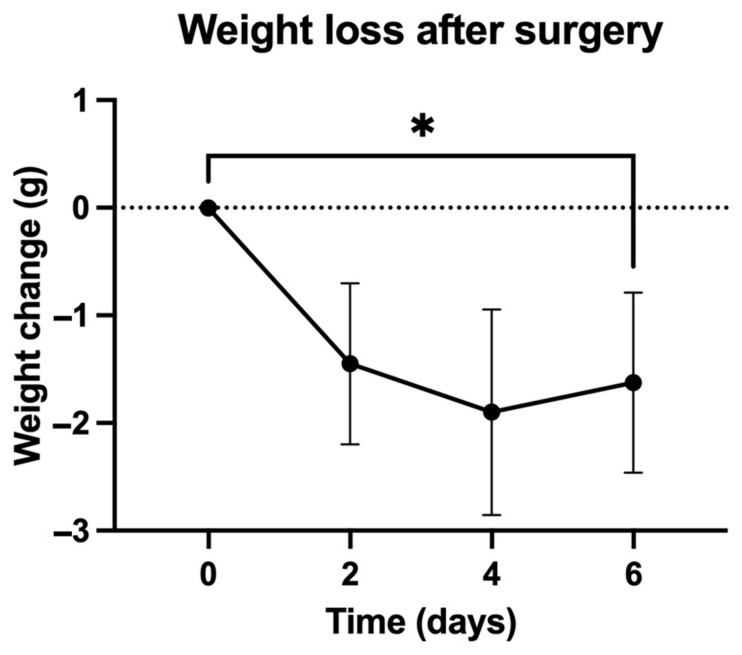
Weight variation (grams) in mice after surgery (* *p* < 0.001). Data are presented as mean ± SD.

**Figure 2 antibiotics-13-01225-f002:**
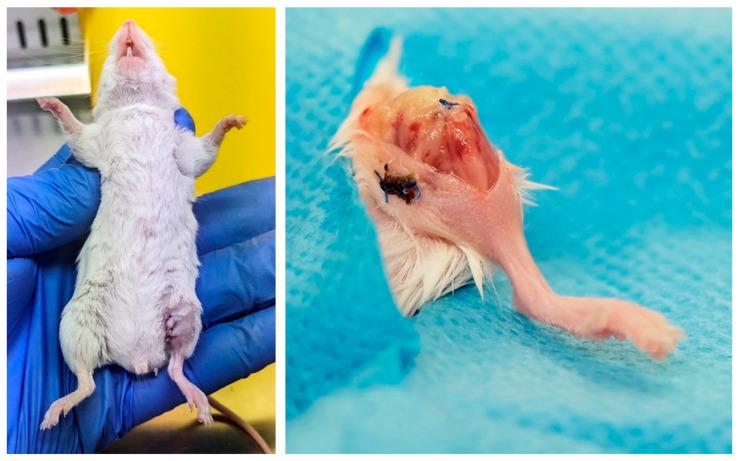
Appearance after surgery. **Left:** Appearance of the wound 7 days after implantation. **Right:** Abscess in the knee joint and subcutaneous tissue observed in all mice in the treatment groups (infected implants).

**Figure 3 antibiotics-13-01225-f003:**
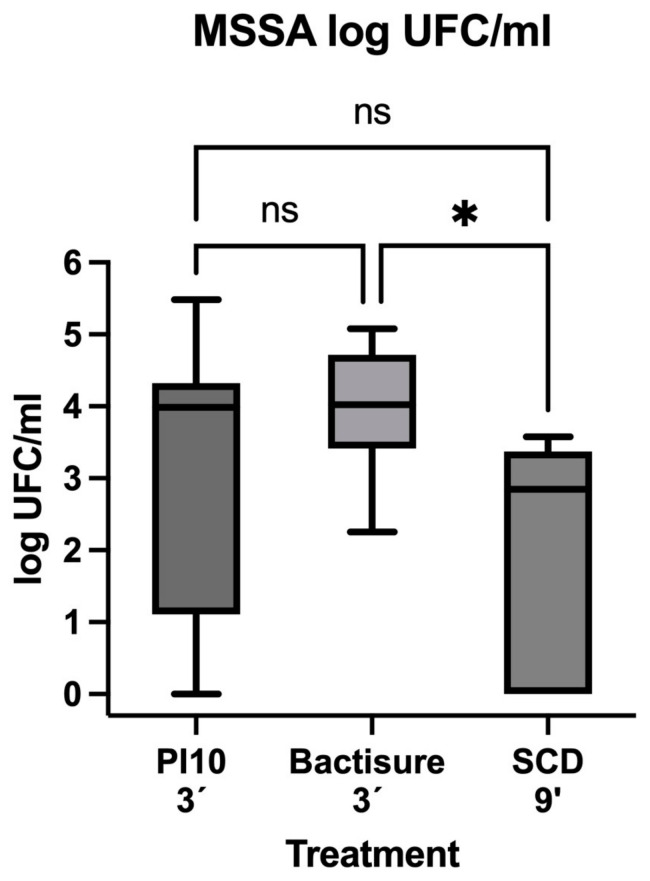
Comparison of the results in terms of median log CFU/mL for each treatment. PI: povidone iodine; SCD: sequential combination debridement; CFU: colony-forming units; ns: non-significant; * *p* = 0.03.

**Figure 4 antibiotics-13-01225-f004:**
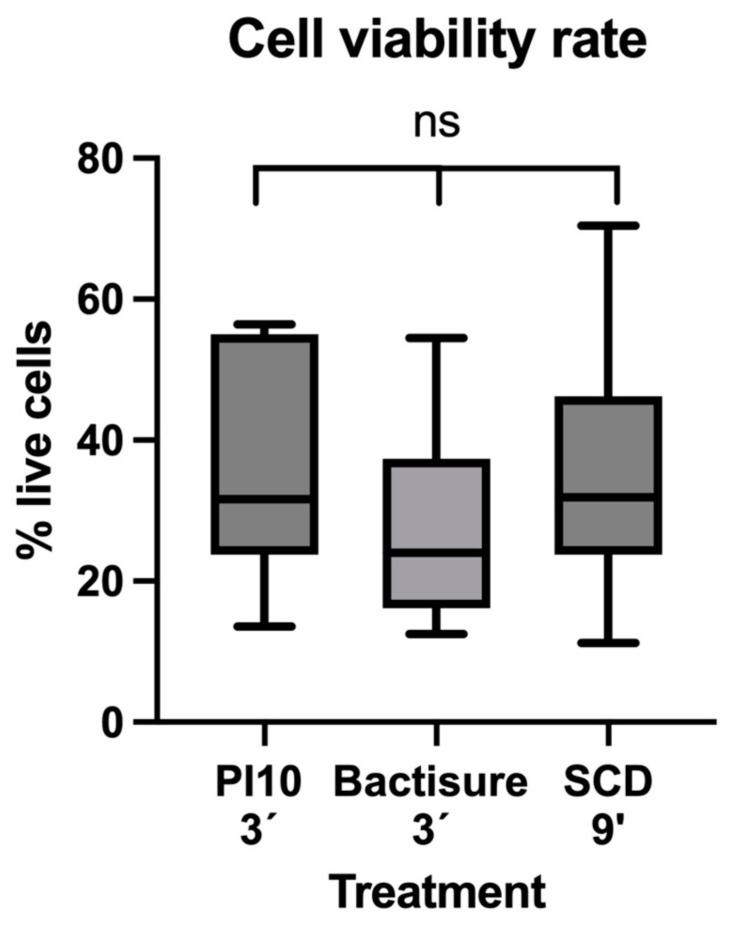
Cell viability rate in each study group. PI: povidone iodine; SCD: sequential combination debridement; ns: non-significant (*p* = 0.34).

**Figure 5 antibiotics-13-01225-f005:**
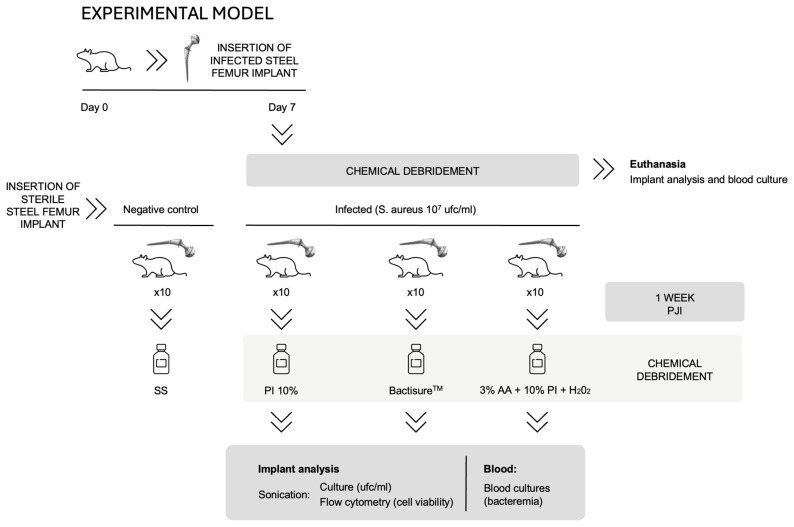
Diagram of the protocol for the murine model of periprosthetic joint infection.

**Figure 6 antibiotics-13-01225-f006:**
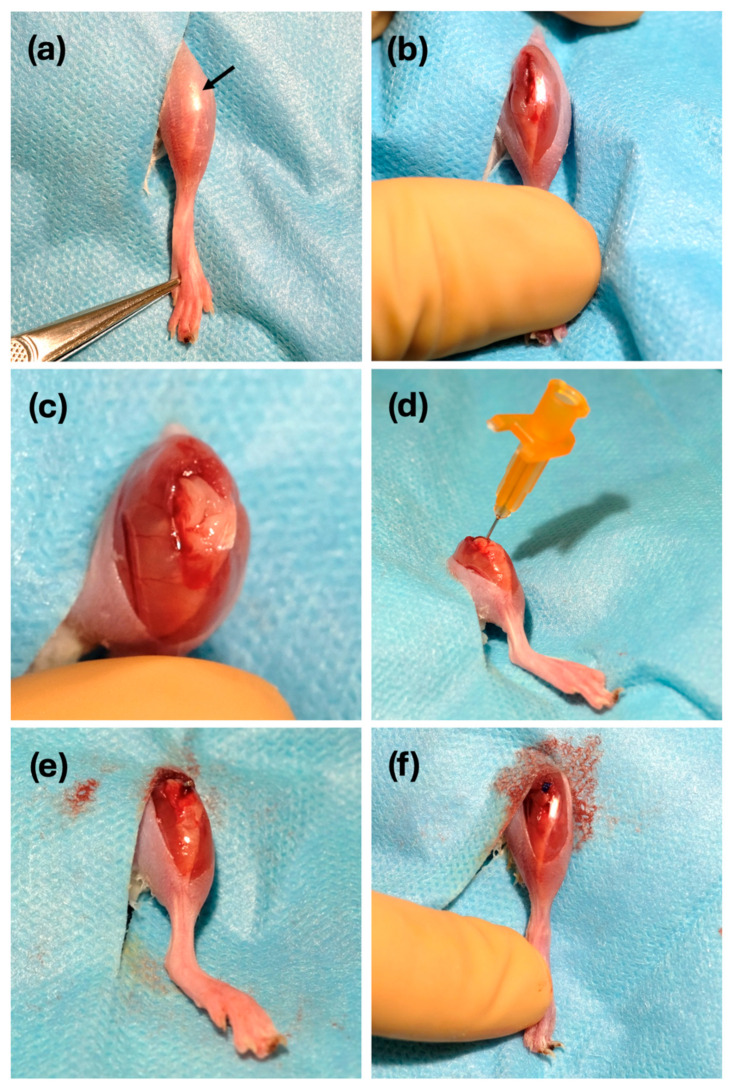
Main steps of implant surgery. (**a**,**b**) Location of the patellar tendon (arrow) and medial parapatellar arthrotomy. (**c**) Luxation of the extensor mechanism and access to the distal femur. (**d**) Creation of the medullary canal with a 25G needle. (**e**) Retrograde insertion of the implant. (**f**) Closure of the arthrotomy and skin with non-absorbable sutures.

**Table 1 antibiotics-13-01225-t001:** Differences in weight loss (grams) between mice in the negative control group (no infection) and mice in the treatment groups.

	Mean (SD) Weight Loss (Grams)		*p*
Control (*n* = 10)	1.60 (1.17)		0.52
PI10 (*n* = 10)	2.20 (1.03)	1.85 (0.88)	
Bactisure™ (*n* = 10)	1.40 (0.52)		
SCD (*n* = 10)	1.20 (0.92)		

PI: povidone iodine; SD: standard deviation; SCD: sequential combination debridement; *p*: *p*-value.

**Table 2 antibiotics-13-01225-t002:** Results in terms of log CFU/mL according to study group.

	CFU/mL	log CFU/mL	Median (IQR) log CFU/mL	*p* *
Control	0	0.00	0.00 (0.00–0.00)	NA
	0	0.00		
	0	0.00		
	0	0.00		
	0	0.00		
	0	0.00		
	0	0.00		
	0	0.00		
	0	0.00		
	0	0.00		
PI10	15,000	4.18	3.99 (1.11–4.33)	**0.017**
	1750	3.24		
	10,000	4.00		
	304,000	5.48		
	30	1.48		
	0	0.00		
	9500	3.98		
	19,000	4.28		
	29,000	4.46		
	0	0.00		
Bactisure™	1520	3.18	4.02 (3.41–4.72)	
	120,000	5.08		
	60,000	4.78		
	16,000	4.20		
	3100	3.49		
	47,000	4.67		
	180	2.26		
	50,000	4.70		
	6910	3.84		
	3080	3.49		
SCD	2960	3.47	2.85 (0.00–3.72)	
	450	2.65		
	1100	3.04		
	0	0.00		
	300	2.48		
	1220	3.09		
	3760	3.58		
	0	0.00		
	0	0.00		
	2180	3.34		

PI: povidone iodine; IQR: interquartile range; CFU: colony-forming units; SCD: sequential combination debridement; NA: not applicable; *p*: *p*-value. * Values in bold indicate statistical significance.

**Table 3 antibiotics-13-01225-t003:** Post hoc analysis comparing treatment groups in terms of log CFU/mL.

	Treatment	Median (IQR) log CFU/mL	*p* *
PI10 vs. Bactisure™	PI10 3′	3.99 (1.11–4.33)	0.99
	Bactisure™ 3′	4.02 (3.41–4.72)	
PI10 vs. SCD	PI10 3′	3.99 (1.11–4.33)	0.19
	SCD	2.85 (0.00–3.72)	
Bactisure™ vs. SCD	Bactisure™ 3′	4.02 (3.41–4.72)	**0.02**
	SCD	2.85 (0.00–3.72)	

PI: povidone iodine; SCD: sequential combination debridement; IQR: interquartile range; CFU: colony-forming units; *p*: *p*-value. * Values in bold indicate statistical significance.

**Table 4 antibiotics-13-01225-t004:** Bacteremia rate in each study group.

	Bacteremia, N (%)	*p* *
Control (*n* = 10)	0 (0.0)	**0.036**
PI10 (*n* = 10)	3 (30.0)	
Bactisure™ (*n* = 10)	1 (10.0)	
SCD (*n* = 10)	1 (10.0)	

SD: standard deviation; SCD: sequential combination debridement; *p*: *p*-value. * Values in bold indicate statistical significance.

## Data Availability

Data are contained within the article and Appendix A.

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
