# Peer review of "Does a Specific Sequential Combination of Antiseptic Solutions for Chemical Debridement in Periprosthetic Joint Infection Improve Outcomes vs. Solution Alone? An In Vivo Study"

_antibiotics, 2024, doi:10.3390/antibiotics13121225_

Round 1
Reviewer 1 Report
Comments and Suggestions for Authors
The manuscript investigates the efficacy of a specific sequential combination of 3 antiseptic solutions (SCD) for chemical debridement in the treatment of periprosthetic joint infections (PJI) and compares it to single treatments with 10% povidone-iodine (PI) and Bactisure™. The study offers a promising basis for exploring the potential benefits of SCD in reducing bacterial load and biofilms, particularly as observed in an in vivo mouse model.
The manuscript is well organised and clear. The abstract summarises the study well, but could better highlight its relevance to clinical practice. The methods are detailed and easy to follow, although some decisions, such as the short biofilm model, could do with more explanation. The results are well presented with clear tables and figures, but could be better linked to the discussion.
The discussion links the results to some previous studies and mentions the study's limitations, but it could focus more on how the findings are relevant to clinical practice. There is also a lack of connection to many other studies that have looked at similar topics. It would be helpful to include a discussion about the ICM 2018 guidelines, which focus on dilution (e.g 9 liters) instead of the types of solutions used.
Line 46: Please explain the biofilm concept in a few words/sentences as it is very important for your study. In contrast, other parts of the introduction could be deleted or shortened.
Line 88: Although it is mentioned in the limitations, it raises the question of why 7 days were chosen. The DAIR procedure assumes that a mature biofilm develops after 14–21 days. The study could have avoided this issue by choosing a longer time frame. Please explain why 7 days were selected.
Line 289-292: These lines are confusing in the discussion. Please remove them here. This data should be mentioned in the Results section instead.
Please mention in the limitations that these statements can only be applied to Staphylococcus aureus.
Author Response
- The abstract summarises the study well, but could better highlight its relevance to clinical practice.
We have revised the abstract to better highlight how our findings translate into clinical practice, particularly emphasizing the potential for integrating our sequential combination (SCD) of antiseptics (3% acetic acid [AA], 10% povidone-iodine [PI], and 5 mM hydrogen peroxide [H2O2]) into standardized protocols for chemical debridement during periprosthetic joint infection (PJI) surgery.
- The methods are detailed and easy to follow, although some decisions, such as the short biofilm model, could do with more explanation.
We chose the 24-hour biofilm model to simulate early infection stages, which is clinically relevant for immediate interventions such as DAIR (debridement, antibiotics, irrigation, and retention). This timeframe aligns with studies exploring immature biofilms, which are more susceptible to treatment. We added this explanation in the Methods section.
- The results are well presented with clear tables and figures, but could be better linked to the discussion.
We have added further references to the results throughout the discussion, emphasizing their relevance and integrating them into the context of our findings. Additionally, we have elaborated on the clinical impact of the study's outcomes, particularly regarding the potential application of sequential combination debridement (SCD) protocols in surgical practice to enhance biofilm eradication and infection control. This ensures a clearer connection between our results and their implications.
- The discussion links the results to some previous studies and mentions the study's limitations, but it could focus more on how the findings are relevant to clinical practice. There is also a lack of connection to many other studies that have looked at similar topics. It would be helpful to include a discussion about the ICM 2018 guidelines, which focus on dilution (e.g 9 liters) instead of the types of solutions used.
We have added a paragraph providing a broader mechanistic explanation of how the combination of 3% AA, 10% PI, and 5 mM Hâ‚‚Oâ‚‚ achieves a synergistic effect, emphasizing its ability to reduce biofilm viability and strengthening the theoretical foundation for SCD. This approach is relevant not only to orthopaedic surgery but also to implant-related infections across various surgical fields.
Dilution is distinct from chemical debridement. According to the ICM 2018 (Blom A, Cho J, Fleischman A, Goswami K, Ketonis C, Kunutsor SK, et al. General Assembly, Prevention, Antiseptic Irrigation Solution: Proceedings of International Consensus on Orthopedic Infections. The Journal of Arthroplasty. 2019 Feb;34(2):S131–8.), the optimal irrigation solution for clean elective orthopedic procedures is dilute PI, although the optimal volume remains undefined. However, no specific recommendation is made for irrigation during PJI surgery due to insufficient evidence. While dilution with sterile saline can reduce bacterial load, antiseptics play a crucial role in eradicating infection by disrupting biofilm and creating a hostile environment that prevents future bacterial growth.
- Line 46: Please explain the biofilm concept in a few words/sentences as it is very important for your study. In contrast, other parts of the introduction could be deleted or shortened.
We have added a brief explanation of the biofilm concept in the introduction, highlighting its structured bacterial communities encased in a protective matrix that confer resistance to antibiotics and antiseptics. Other parts of the introduction were shortened to maintain focus.
- Line 88: Although it is mentioned in the limitations, it raises the question of why 7 days were chosen. The DAIR procedure assumes that a mature biofilm develops after 14–21 days. The study could have avoided this issue by choosing a longer time frame. Please explain why 7 days were selected.
The 7-day infection model was selected based on previously validated studies (Lovati AB, Drago L, Monti L, De Vecchi E, Previdi S, Banfi G, et al. Diabetic Mouse Model of Orthopaedic Implant-Related Staphylococcus aureus Infection. Beloin C, editor. PLoS ONE. 2013 Jun 20;8(6):e67628.) and to balance infection development with ethical considerations while minimizing distress to the mice. Extending the model to 14–21 days presents challenges in maintaining infection consistency and ensuring animal health. This has been explicitly clarified in the Methods section.
- Line 289-292: These lines are confusing in the discussion. Please remove them here. This data should be mentioned in the Results section instead.
We agree that these lines are not suited for the discussion section. As the data originates from a previous study, we reference the results without including detailed data and opted not to include it in the Results section to avoid redundancy and maintain focus on the current study's findings.
- Please mention in the limitations that these statements can only be applied to Staphylococcus aureus.
We have clearly stated in the limitations that our findings are specific to Staphylococcus aureus and cannot be directly extrapolated to other organisms or polymicrobial biofilms.
Reviewer 2 Report
Comments and Suggestions for Authors
This study is an interesting contribution to understanding the efficacy of different approaches for managing biofilms in periprosthetic joint infections (PJI). The in vivo design and the detailed methodology described in the manuscript are commendable. However, a few aspects invite further discussion.
Was the choice to use Staphylococcus aureus as the sole test organism based on its clinical relevance, and might future studies benefit from exploring polymicrobial biofilms ? how could using different microorganisms stand in relation to the results you obtained now? Can you anticipate different results?
The inter-animal variability appears to be very high in certain conditions. Is that to be expected with this model? It should probably discussed in further detail
Author Response
- Was the choice to use Staphylococcus aureus as the sole test organism based on its clinical relevance, and might future studies benefit from exploring polymicrobial biofilms? how could using different microorganisms stand in relation to the results you obtained now? Can you anticipate different results?
Staphylococcus aureus was selected due to its clinical relevance in PJI and its strong biofilm-forming capacity. We agree that future studies should investigate polymicrobial biofilms, as their structural complexity and interspecies interactions may significantly influence treatment efficacy. These potential variations in antiseptic performance have been acknowledged in the limitations.
- The inter-animal variability appears to be very high in certain conditions. Is that to be expected with this model? It should probably be discussed in further detail
The reviewer is absolutely correct. Variability among animals in the in vivo model is expected due to host-specific factors such as immune response and biofilm formation dynamics. This limitation has been addressed in greater detail, highlighting its potential impact on the statistical power of our results.
Reviewer 3 Report
Comments and Suggestions for Authors
This paper, “Do specific sequential combinations of antimicrobial solutions for chemical debridement for periprosthetic infection work better than either solution alone? An in vivo study”, explores an important topic in the field of orthopedics and infection management. The efficacy of sequential chemical debridement (SCD) is validated using a mouse model, adding value to previous in vitro studies.
Is Bactisuret TM a mixture? Please clarify. Can you elaborate on the rationale for the selection of antimicrobial solution concentrations and their sequence. Are these optimized for in vivo use or are they directly extrapolated from the in vitro findings?
This study used K-wires instead of screws. K-wires are more similar to nails and their use suggests that this model may better represent fracture-related infections rather than periprosthetic joint infections
Why did the authors not analyze adjacent bone tissue? Assessment of bacterial load or biofilm presence in the surrounding bone tissue would provide more insight into infection dynamics and efficacy of antimicrobial agents.
The mice were euthanized immediately after debridement. Can the observed bacteremia be attributed to the debridement procedure in such a short period of time or is it simply related to pre-existing infection? This point requires further clarification and discussion.
Tables 2 and 5 appear overly complex. Simplifying them to numbers or graphical representations (e.g., bar graphs or box plots) would make the data easier to understand and more visually clear for the reader.
The authors conclude that SCD is the most effective method for reducing bacterial load. However, this conclusion is not supported by statistically significant results in all comparisons. The statement should be revised to reflect the limitations of the statistical results and avoid overgeneralization.
The data mentioned in lines 289-291 are not clearly cited. Please state the source of these results and add appropriate citations or supplementary information as needed.
Provide more mechanistic explanations on how the combination of 3% AA, 10% PI, and 5 mM Hâ‚‚Oâ‚‚ achieves a synergistic effect. This could strengthen the theoretical basis for SCD.
Suggest that extended time points are needed to evaluate the long-term effects of debridement on bacteremia and infection control. A more extensive discussion of how these findings may impact clinical protocols would enhance the impact of the manuscript.
Author Response
- Is Bactisuret TM a mixture? Please clarify. Can you elaborate on the rationale for the selection of antimicrobial solution concentrations and their sequence. Are these optimized for in vivo use or are they directly extrapolated from the in vitro findings?
Bactisure™ is a commercial mixture containing ethanol, acetic acid, sodium acetate, and benzalkonium chloride. Concentrations for antiseptics were chosen based on their demonstrated efficacy in prior in vitro studies (Márquez-Gómez M, Díaz-Navarro M, Visedo A, Hafian R, Matas J, Muñoz P, et al. An In Vitro Study to Assess the Best Strategy for the Chemical Debridement of Periprosthetic Joint Infection. Antibiotics. 2023 Oct 2;12(10):1507.).
The sequence of application (acetic acid, povidone-iodine, hydrogen peroxide) was optimized to exploit the disruptive effects of acetic acid on the biofilm matrix, followed by the bactericidal action of povidone-iodine and hydrogen peroxide, as we observed in our previous study.
- This study used K-wires instead of screws. K-wires are more similar to nails and their use suggests that this model may better represent fracture-related infections rather than periprosthetic joint infections
While K-wires were used in this study, the implant was introduced into the medullary canal of the femur, leaving 1 mm distal intra-articular. This design choice was made to better simulate the scenario of a knee PJI, where the implant typically includes intra-articular components.
- Why did the authors not analyze adjacent bone tissue? Assessment of bacterial load or biofilm presence in the surrounding bone tissue would provide more insight into infection dynamics and efficacy of antimicrobial agents.
We did not assess bacterial load in adjacent bone tissue due to the short infection duration (7 days), which likely limited biofilm spread to surrounding tissue. This has been noted as a limitation, and we recommend extended-duration studies to comprehensively evaluate infection dynamics.
- The mice were euthanized immediately after debridement. Can the observed bacteremia be attributed to the debridement procedure in such a short period of time or is it simply related to pre-existing infection? This point requires further clarification and discussion.
Bacteremia observed immediately after debridement likely reflects a pre-existing infection due to the rapidity of the observed effect. This interpretation has been clarified in the discussion.
- Tables 2 and 5 appear overly complex. Simplifying them to numbers or graphical representations (e.g., bar graphs or box plots) would make the data easier to understand and more visually clear for the reader.
Table 2 is already presented as a box plot in Figure 5. We have removed Table 5 and replaced it with a simplified graphical representation (see Figure 6).
- The authors conclude that SCD is the most effective method for reducing bacterial load. However, this conclusion is not supported by statistically significant results in all comparisons. The statement should be revised to reflect the limitations of the statistical results and avoid overgeneralization.
We have revised our conclusions to ensure they are proportionate to the data presented, avoiding overgeneralization in light of the lack of statistical significance in certain comparisons.
- The data mentioned in lines 289-291 are not clearly cited. Please state the source of these results and add appropriate citations or supplementary information as needed.
This has already been clarified. Please refer to point 7 from reviewer 1.
- Provide more mechanistic explanations on how the combination of 3% AA, 10% PI, and 5 mM Hâ‚‚Oâ‚‚ achieves a synergistic effect. This could strengthen the theoretical basis for SCD.
We have expanded the discussion to elaborate on the synergistic mechanism, detailing how AA disrupts the biofilm matrix, thereby enabling enhanced penetration and bactericidal effects of PI and Hâ‚‚Oâ‚‚. As a weak organic acid, AA’s diffusion capacity allows it to reach tissue regions that are inaccessible to mechanical debridement or other antiseptics with lower penetration abilities (Williams RL, Ayre WN, Khan WS, Mehta A, Morgan-Jones R. Acetic Acid as Part of a Debridement Protocol During Revision Total Knee Arthroplasty. The Journal of Arthroplasty. 2017 Mar 1;32(3):953–7.).
While AA is not primarily used for its antiseptic properties due to its prolonged action time (approximately 20 minutes) to achieve bactericidal effects (Tsang STJ, Gwynne PJ, Gallagher MP, Simpson AHRW. The biofilm eradication activity of acetic acid in the management of periprosthetic joint infection. Bone & Joint Research. 2018 Aug 1;7(8):517–23.), its role in combination therapy is critical. By breaking down the biofilm matrix, AA facilitates the subsequent bactericidal action of PI, which directly targets bacteria, and Hâ‚‚Oâ‚‚, which increases cell wall porosity and enhances the effectiveness of PI due to their synergistic interaction.
- Suggest that extended time points are needed to evaluate the long-term effects of debridement on bacteremia and infection control. A more extensive discussion of how these findings may impact clinical protocols would enhance the impact of the manuscript.
Future studies should assess the long-term impacts of chemical debridement on infection control and bacteremia. However, evaluating these effects in animal models is challenging due to the difficulty of maintaining the infection model over extended periods. Additionally, confirming the safety profile of this protocol is essential, even though these solutions are routinely used in clinical practice without an observed increase in surgical wound complications. This need has been added in the discussion.
Round 2
Reviewer 2 Report
Comments and Suggestions for Authors
I think the authors have addressed the concerns I raised